# The Development of a Measuring System for Intraoral SpO_2_

**DOI:** 10.3390/s24020435

**Published:** 2024-01-10

**Authors:** Yuki Kashima, Minako Onimaru, Ryosuke Isogai, Noboru Kawai, Yoshifumi Yoshida, Koutaro Maki

**Affiliations:** 1Department of Orthodontics, School of Dentistry, Showa University, 2-1-1 Kitasenzoku, Ota-ku, Tokyo 145-8515, Japan; onimaru@dent.showa-u.ac.jp (M.O.); maki@dent.showa-u.ac.jp (K.M.); 2Research and Development Department, Seiko Future Creation Inc., 563, Takatsuka Shinden, Chiba 270-2222, Japanyoshifumi.yoshida@seiko-sfc.co.jp (Y.Y.)

**Keywords:** pulse oximetry, mouthpiece, pulse rate, oxygen saturation monitoring, wearable monitoring system, SpO_2_

## Abstract

Blood oxygen saturation (SpO_2_) is an essential indicator of a patient’s general condition. However, conventional measurement methods have some issues such as time delay and interference by ambient light. Improved measurement methods must be developed, and there are no reports on intraoral measurements of SpO_2_ using wearable devices. Therefore, we aimed to establish an intraoral SpO_2_ measurement method for the first time. Twelve healthy adults participated in this study. The following steps were taken: (1) to identify the optimal measurement location, mid-perfusion index (PI) values were measured at six places on the mucosa of the maxilla, (2) to validate the optimal measurement pressure, PI values were obtained at different pressures, and (3) using the proposed mouthpiece device, SpO_2_ values in the oral cavity and on the finger were analyzed during breath-holding. The highest PI values were observed in the palatal gingiva of the maxillary canine teeth, with high PI values at pressures ranging from 0.3 to 0.8 N. In addition, changes in SpO_2_ were detected approximately 7 s faster in the oral cavity than those on the finger, which is attributed to their proximity to the heart. This study demonstrates the advantage of the oral cavity for acquiring biological information using a novel device.

## 1. Introduction

In recent years, the acquisition and management of biometric information using wearable devices has become common. Blood oxygen saturation (SpO_2_), indicating the oxygen supply to the human body, is one of the main items detected by wearable devices. It is indispensable for monitoring vital organs such as the heart and brain. SpO_2_ measurements are currently used in many situations, such as checking vital signs during exercise, monitoring the condition of older patients, and monitoring sick and hospitalized patients [1,2].

There are two types of SpO_2_ sensors: transmissive sensors that are attached to the fingers and toes and reflective sensors that are fixed by applying light pressure to the forehead or other parts of the body [3]. Transmissive sensors use a light source and a detector positioned across the measurement site or living tissue. Their advantages include rapid use and the ability to easily sample various parts of the body, even with low-amplitude signals. Reflective sensors use the light source and detector positioned on the same side of the living tissue. In cold environments, human arteries constrict to preserve body heat, minimizing heat loss. Consequently, in transmission-type devices, a signal decline from arterial capillaries impairs measurement accuracy. This issue is avoided in reflection-type pulse oximeters, where the sensor (comprising LEDs and a photodetector) can be positioned near any body part [4,5]. In general, transmissive SpO_2_ sensors worn on the hand are widely used because they are considered to be easy to install and use. However, they have six major problems as outlined below.

Time delay in measurements: There is a time delay associated with measurements taken at the periphery of the body, such as the fingers, which makes it difficult to detect sudden changes in pathological conditions in real time [6].Dependence on blood flow: When blood flow decreases because of insufficient peripheral circulation, such as during hypothermia, it is difficult to obtain sufficiently strong signals [7].The influence of body movement: Body movement during sports or sleep causes the device to shift, which significantly affects the measurement accuracy [8].Location: Many conventional devices are worn on the surface of the body, which is difficult to accomplish in cases where the patient has burns or sensitive injuries [9].The influence of pressure: Accurate measurement is impossible without using the optimal contact pressure [10]. If the pressure of the sensor on the human body is insufficient, the amplitude of the AC signal is reduced, making it difficult to detect transmitted light. Conversely, when excessive pressure is applied, the waveform of the AC signal is distorted by the occluded artery, which also affects the accuracy [11].The influence of visible light: Interference from visible light in the surroundings can affect the accurate detection of the desired light signals [12].

To overcome the shortcomings of conventional pulse oximeters, several studies have reported on the measurements of SpO_2_ at various sites, such as the external auditory canal [13], intestine [14], and esophagus [15]. However, no previous reports have completely solved the problems mentioned above. Therefore, it remains necessary to establish a measurement method at a site where stable and accurate measurements can be obtained.

Herein, we propose a measurement method that solves the above problems by mounting the device on a mouthpiece, referred to as an intraoral device. Since the oral cavity is close to the heart, there is little time delay in the SpO_2_ measurement, and blood flow is always abundant. The mouth is also dark and is not affected by ambient light to a significant extent. In addition, SpO_2_ is conventionally measured through the epidermis, dermis, and subcutaneous tissue, using a skin-mediated method. However, there are more blood vessels in the oral mucosa than in the skin [16,17]. Therefore, blood flow is not only abundant and easy to measure, but the measured values may also represent the blood oxygen concentrations inside the body more accurately.

Furthermore, mouthpieces are widely used for orthodontic treatment and can be easily put on and taken off [18]. Thus, an example of its application in intraoral sensing has been reported [19,20]. Notably, this type of device is unlikely to shift because of body movement, because it is stabilized within the oral cavity. However, the oral cavity is a moist environment, making it necessary to consider waterproofing. Additionally, conventional transmissive pulse oximeters cannot be used because biological tissue cannot be easily confined in the measuring instrument, and the effects of pressure are unknown. For these reasons, measuring SpO_2_ in the oral cavity has been challenging.

Therefore, we developed a waterproof reflective SpO_2_ sensor, which represents the first intraoral SpO_2_ measurement method using a mouthpiece-type device. We also verified the optimal measurement site and pressure.

## 2. Materials and Methods

### 2.1. Principles of Measurement

#### 2.1.1. Pulse Oximeter

The pulse oximeter is based on the principle that oxygenated hemoglobin molecules (HbO_2_) and deoxygenated hemoglobin molecules (Hb) absorb different amounts of red and near-infrared (IR) light [3]; HbO_2_ absorbs a larger amount of IR light and a smaller amount of red light than Hb. Pulse oximeters emit two wavelengths of light from a small light-emitting diode: generally red light at 660 nm and near-IR light at 940 nm. In a transmission-type device, the light transmitted through the sample is detected using a photodiode (PD) placed on the opposite side. In a reflection-type device, the PD detects the reflected light from the body tissue [11].

#### 2.1.2. Perfusion Index (PI) Value

The detected optical signal can be separated into a pulsatile (AC) signal that fluctuates with the pulse wave and a non-pulsatile (DC) signal that does not fluctuate (Figure 1). The AC/DC signal ratio expressed is called the perfusion index (PI) value. The PI value is used in the field of pulse oximetry as an indicator of measurement stability [21] and indicates the immediate perfusion state of the tissue in the area of application at regular time intervals. Higher PI values indicate a stable measurement because of the state of high tissue perfusion. When the PI value is small, the state of tissue perfusion is low, making the measurement susceptible to noise and reducing the accuracy of SpO_2_ measurements [22].

### 2.2. Intraoral Sensing Modules

#### 2.2.1. Sensing Module and Measurement System

In this study, the measurement system consists of sensing modules (SpO_2_ and pressure sensors), an external control board, and a computer. Figure 2 shows a block diagram of the measurement system. The SpO_2_ sensor consists of two light-emitting diodes (LEDs) (660 and 880 nm, MAXM86161, Analog Devices, Cambridge, MA, USA). The measurement condition of SpO_2_ is listed in Table 1. The pressure sensor (HSFPAR303A, Alps Alpine Co., Ltd., Tokyo, Japan) is a load sensor with a piezoresistor formed on a silicon diaphragm. This sensor utilizes the piezo-resistive effect, in which stress is generated in the piezoresistor by the flexure of the diaphragm when a load is applied, resulting in a change in resistivity. In addition, the pressure sensor is given a protrusion to enable more accurate pressure measurements compared with surface-type pressure sensors. Figure 3 shows a sensing module and its design diagram. The module consists of an optical sensor and a pressure sensor, which measure light intensity and SpO_2,_ while also measuring the mechanical pressure (Figure 3a,b). The compact design of the module (SpO_2_ sensor: height of 6.5 mm, width of 8.0 mm, depth of 3.0 mm; pressure sensor: height of 6.5 mm, width of 8.0 mm, depth of 2.06 mm) makes it easy to follow the individual’s palate (Figure 3d,e). Furthermore, the wiring between the sensor and connector was formed using a flexible printed circuit, allowing installation from the mouth to any intraoral location. The sensing module is connected to an external control board by a cable and controlled using a microcontroller via a twin-wire interface. Figure 4 shows the waterproof structure of the sensor. The sensor is coated with parylene and silicone to provide a double waterproof structure. However, the SpO_2_ sensor is exposed to prevent the optical absorption of silicone.

#### 2.2.2. Method of Manufacturing the Mouthpiece Device

Mouthpieces equipped with SpO_2_ sensors need to be custom-made for each subject. To do this, first, optical impressions of the maxilla were taken (TRIOS, 3Shape, Copenhagen, Denmark) and reconstructed using a 3D printer (AGILISTA, KEYENCE, Osaka, Japan) to create a mold. An 8.0 mm × 6.5 mm × 1.0 mm sensor storage spacer was bonded to the palatal gingival portion of maxillary #3, and a 0.80 mm mouthpiece sheet material (Erkodur; Erkodent Erich Kopp GmbH, Pfalzgrafenweiler, Germany) was placed in a vacuum thermoforming unit (Erkopress ci motion; Erkodent Erich Kopp GmbH) and thermoformed. Then, the mouthpiece was removed from the dental model and modified again (Figure 5).

### 2.3. Measurement of SpO_2_

#### 2.3.1. Verification of the Mouthpiece Device (Extraoral)

The verification equipment is capable of verifying a transmission-type pulse oximeter. Therefore, we decided to verify the accuracy by employing a pulse oximeter whose accuracy was verified by the verification device.

We evaluated the accuracy of the pulse oximeter (PVM-2701, Nihon Kohden, Tokyo, Japan) in advance using the validation device (vPad-A1, Metts, Tokyo, Japan) and confirmed that there were no problems with accuracy. The pseudo-signal of light transmitted through living tissue, such as a fingertip, emitted from the verification device was read by the pulse oximeter. We verified whether the value was the same as the vital sign of the validation device.

The PVM-2701 pulse oximeter was then attached to the middle finger of the right hand, while the developed SpO_2_ sensor was grasped with the index finger of the right hand so that it contacted the palm surface. SpO_2_ values were simultaneously monitored using the two sensors for 30 s (Figure 6). Bland–Altman analysis was performed to determine the agreement between the developed SpO_2_ sensor and the commercial pulse oximeter [23,24]. JMP Pro^®^ 16.0.0 software (SAS Institute Inc., Cary, NC, USA) was used for all statistical analyses.

#### 2.3.2. Accuracy Evaluation of the Mouthpiece Device (Intraoral and Extraoral)

The developed SpO_2_ sensor was used to monitor the oral cavity and finger at rest for 30 s. The developed SpO_2_ sensor was attached to a mouthpiece and placed in the oral cavity (Figure 7), while another device was held with the index finger of the right hand so that the developed SpO_2_ sensor was in contact with the palm surface. Bland–Altman analysis was performed to determine the agreement between the two monitoring sites.

#### 2.3.3. SpO_2_ and Pressure Measurements at Different Locations (Intraoral)

Using devices constructed for six healthy adult volunteers, SpO_2_ was measured at the six intraoral locations shown in Figure 8 while changing the contact pressure. The PI values were also obtained.

#### 2.3.4. Breath-Holding Measurements Using the Mouthpiece Device (Intraoral and Extraoral)

Twelve healthy adult volunteers participated in this study. They had normal occlusion and no history of oral or maxillofacial disease or extreme trauma. All subjects were seated in a chair at rest. The mouthpiece device was placed in the oral cavity while another device was held with the index finger of the right hand so that the developed SpO_2_ sensor was in contact with the palm surface. The device used for measurements was the same for both the oral cavity and the fingers. After normal breathing was continued from the start of measurement, breath-holding was performed from the state of exhaled breath. After holding the breath within a comfortable range, the subject resumed breathing. Once the SpO_2_ value recovered, the measurement was terminated. Breath-holding was performed to simulate a hypoxic condition conveniently [25]. Measurements were taken three times per subject. The normality of the SpO_2_ values obtained from the oral cavities and fingers based on their lowest point, was verified using the Shapiro–Wilk test. For normally distributed values, a paired *t*-test was used, and for non-normally distributed values, a Wilcoxon signed-rank test was used.

## 3. Results

### 3.1. Accuracy of the Mouthpiece Device (Extraoral)

Bland–Altman analysis was performed to compare the corresponding SpO_2_ values obtained from the developed SpO_2_ sensor and commercial pulse oximeter. There was no significant difference between the measured values and the mean. The limits of agreement were 0.928 to −0.124. Therefore, the developed SpO_2_ sensor and pulse oximeter obtained similar measurements, and there were no problems with the accuracy of the developed SpO_2_ sensor.

### 3.2. Accuracy of the Mouthpiece Device (Intraoral and Extraoral)

From the Bland–Altman analysis, SpO_2_ values obtained from the finger were compared with those obtained from the mouth using the developed SpO_2_ sensor. There was no significant difference between the measured values and the mean. The limits of agreement were 0.241 to −0.694. Thus, there was no significant difference between the SpO_2_ values obtained from the finger and mouth at rest.

### 3.3. Evaluation of Measured Values Obtained Using the Mouthpiece Device

#### 3.3.1. SpO_2_ and Pressure (Intraoral)

The PI values obtained for each measurement site are shown in Figure 9. The PI value for the palatal gingiva of the maxillary canine was the highest, and the PI value for the palatal gingiva of the maxillary premolar was the lowest. The average PI values for all six sites at different pressures are shown in Figure 10. The PI values were relatively high in the pressure range of 0.3–0.8 N.

#### 3.3.2. Breath-Holding and SpO_2_ Response (Intraoral and Extraoral)

An example of the results obtained for the mouth and finger using a mouthpiece SpO_2_ measurement device is shown in Figure 11. The measurements in the oral cavity are more responsive and exhibit lower values than those performed using the finger. During breath-holding, the time(s) of the measurement response for the oral cavity and finger was evaluated. The time during measurement was classified into four segments (T1–T4), as shown in Figure 12. The lowest SpO_2_ value (% of baseline) was also evaluated for the oral cavity and fingers during breath-holding.

Table 2 summarizes the time evaluations (T1–T4) comparing oral cavity and finger measurements during breath-holding. This represents the time difference between the response measured in the oral cavity and the response measured on the finger. All T1–T4 periods showed significant differences (*p* < 0.05), indicating that oral cavity measurements are more responsive than finger measurements. On average, the oral cavity responded about 7 s (18%) earlier than the fingers across T1–T4.

In Table 3, the results for the lowest SpO_2_ values are presented. Significant differences were observed (*p* < 0.05), with the oral cavity showing a lower SpO_2_ value than the finger. The lowest SpO_2_ value measured in the oral cavity showed a 4% difference compared to the value measured on the finger.

## 4. Discussion

Differences in PI values were observed due to different measurement locations in the oral cavity. These differences are attributable to the anatomical structure of the oral cavity, such as the distribution of arteries and blood vessels. Especially on the lingual side of the canine teeth, the major palatine artery runs close to the teeth, and the mucosa is thicker compared to other parts of the mouth [26,27], resulting in more abundant blood flow. Accordingly, the PI value was higher and more stable on the canine palatal side, site 3. The lowest PI values were observed at sites 4 and 5, which may be explained by the thinner mucosa in the molars, where the blood supply was lower. The reason for the low PI value at sites 1 and 2 is considered to be the unevenness of the palatal folds. Even though blood flow is abundant from the nasopalatine canal artery, the curve of the palate may have prevented even contact pressure with the sensor. Although site 6 was located closest to the major palatine artery, the mucosa was thinner than that at site 3, contributing to a lower PI value.

In addition, one of the critical factors for stable sensing is the comfort of the subject, which must be considered to avoid reflective unconscious tongue movement and sensor displacement. To reduce discomfort and avoid tongue interference, the sensor should be set in close proximity to the teeth. In terms of comfort, blood supply, and accurate measurements, the canine teeth are the most appropriate site for measuring SpO_2_ in the oral cavity.

Regarding the effect of pressure, high PI values were observed at 0.3–0.8 N. In a previous study, in which PI values were examined by changing the pressure between the sensor and the measurement site (finger) on the skin, the PI value reached maximum values at 0.2–0.8 N [28]. This result is consistent with the results of the present experiment, despite the differences between skin and mucous membranes. However, a future challenge lies in the potential for pressure fluctuations and instability in measurements due to tongue interference. Therefore, it is essential to exercise caution during measurements to ensure that the tongue does not exert pressure on the sensor.

The contact pressure between the SpO_2_ sensor and the oral cavity varies according to shape, but the shape of the palate differs from person to person and the pressure could not be varied precisely according to shape. The SpO_2_ sensor and pressure sensor were superimposed so that the pressure with which the SpO_2_ sensor contacts the oral cavity can be measured quantitatively. In case of insufficient pressure, the amplitude of the AC signal decreases owing to insufficient contact with the sensor. Conversely, excessive pressure may reduce the amplitude of the AC signal and distort the waveform [11]. Thus, it is necessary to determine the minimum pressure required to accurately measure SpO_2_.

The results obtained during breath-holding suggest that the oral cavity sensor can detect a drop in SpO_2_ approximately 7 s (18%) earlier than the finger sensor, and the lowest SpO_2_ value was approximately 4% more acutely detected in the oral cavity than that obtained on the finger when breath-holding stopped. The reason for this is that the oral cavity is closer to the central vascular system and lungs than the fingers and other terminal organs. Therefore, the time for arterial blood to reach the measurement site is shorter for the mouth than for the finger. Furthermore, compared with peripheral arteries, there is no vasoconstrictive response, and blood with a lower oxygen concentration may reach the mouth more quickly than the finger. The significant decrease in SpO_2_ observed in the oral cavity can be attributed to the proximity of the oral cavity to the lungs compared to the fingers. It is hypothesized that blood with a lower oxygen concentration reaches the measurement site more readily in the oral cavity than in the fingers. A study of in-ear devices, which are placed in the ear like earphones, reported that ear sensors showed faster response than finger sensors [25]. Reflective pulse oximeters worn on the forehead have also been reported to respond more quickly than conventional pulse oximeters [29]. The oral cavity, ears, and forehead are closer to the central vascular system and lungs than the fingers.

## 5. Conclusions

In this study, a small, low-power sensor attached to a mouthpiece was used to measure SpO_2_ in the oral cavity for the first time. The accuracy of the developed SpO_2_ measurement device was verified by comparison with the commercial device. Furthermore, there was no significant difference in the baseline SpO_2_ values obtained between the oral cavity and the finger at rest. In the oral cavity, the glass surface of SpO_2_ sensors without parylene coating became foggy. Subsequently, measurements were taken at different locations in the oral cavity, revealing that the maxillary canine area was the appropriate site for measuring SpO_2_. The pressure range of 0.3 to 0.8 N was determined to be the most appropriate for practical measurements. Moreover, the SpO_2_ values in the oral cavity were reflected approximately 7 s (18%) earlier than those measured on the finger, and changes in SpO_2_ values could be detected more acutely.

Although conventional pulse oximeters are simple to use, measurements at the periphery, such as the fingers, are difficult to obtain when blood flow is reduced because of hypothermia or other circulatory conditions [6,30,31]. Hypothermia is a symptom in older patients and those with multiple diseases and is associated with a decrease in pulse rate as the body temperature falls [32,33]. However, this important sign may be inaccurately measured with a conventional finger-worn pulse oximeter [34,35]. Considering that the oral cavity is close to the center of the body, it should be possible to detect changes in blood oxygen saturation in the central region and a decrease in pulse rate due to hypothermia.

Numerous devices are commonly used in the oral cavity, including dentures, aligners, splints, and functional devices for orthodontic treatment. By adding sensing functions to these devices, more accurate biometric information can be obtained, which can also be used for individual health management and home healthcare. Moreover, important signs that have been impossible to measure with the finger may be detectable in the oral cavity. The present study not only demonstrates the potential of the oral cavity as a new site for acquiring biological information but also establishes a new method for accurately measuring SpO_2_ in the oral cavity.

## Figures and Tables

**Figure 1 sensors-24-00435-f001:**
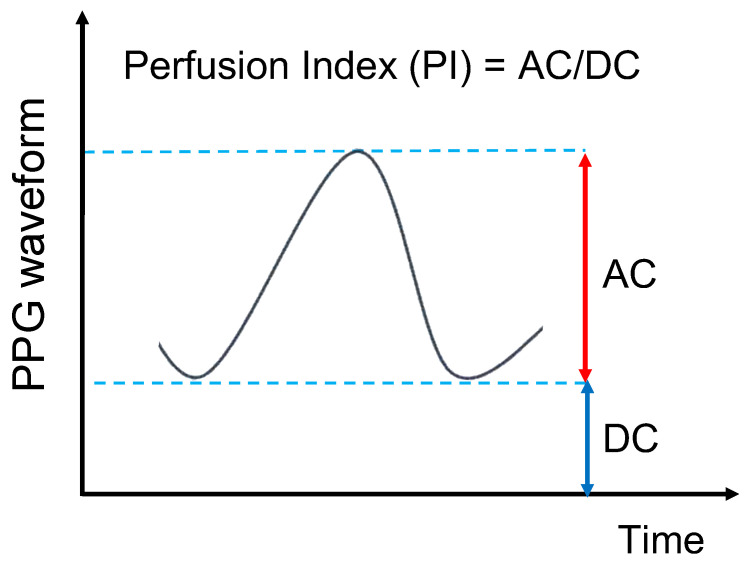
Schematic diagram of AC and DC signals.

**Figure 2 sensors-24-00435-f002:**
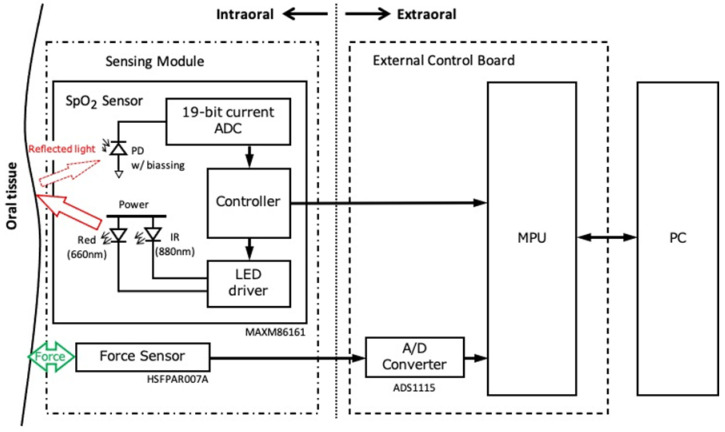
Block diagram of the SpO_2_ measurement system. MPU, microprocessor; PC, personal computer.

**Figure 3 sensors-24-00435-f003:**
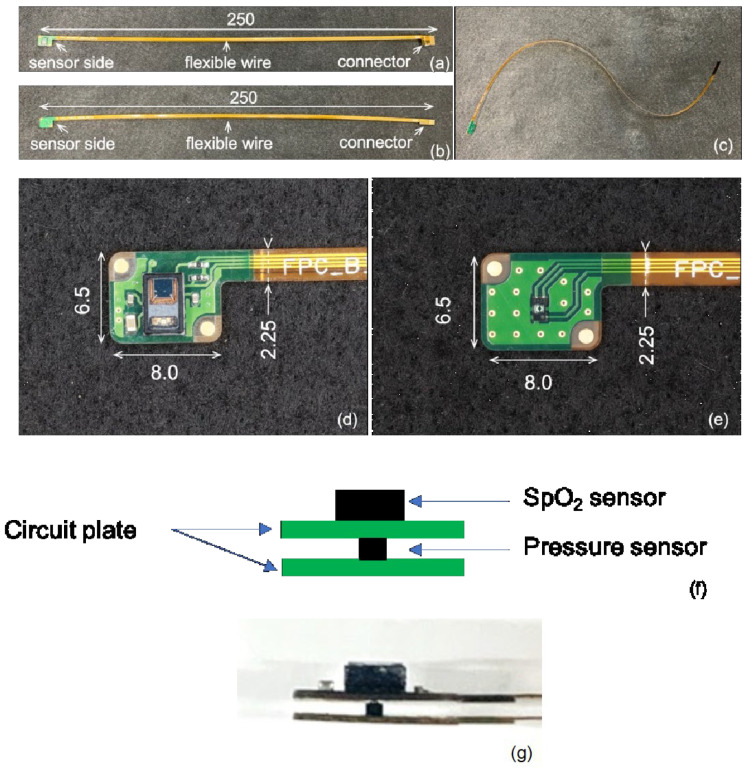
The sensor module. (**a**) Photograph of the SpO_2_ sensor. (**b**) Photograph of the pressure sensor. (**c**) Photograph of sensor wiring in the bent state. (**d**) Magnified view of the SpO_2_ sensor. (**e**) Magnified view of the pressure sensor. (**f**) Schematic diagram of the side view of the SpO_2_ sensor and pressure sensor. (**g**) Side view of the SpO_2_ sensor and pressure sensor.

**Figure 4 sensors-24-00435-f004:**
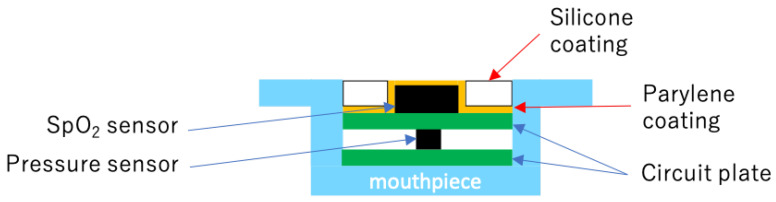
Schematic diagram of the waterproof structure.

**Figure 5 sensors-24-00435-f005:**
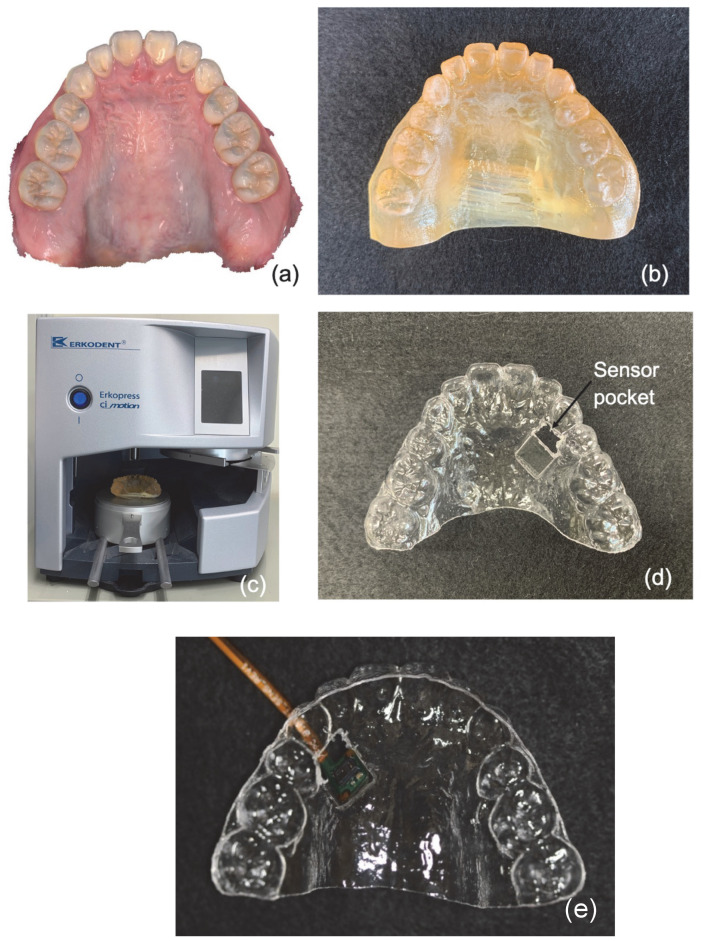
Manufacturing method of the mouthpiece-type SpO_2_ analyzer. (**a**) Optical impression data of the teeth; (**b**) a tooth model fabricated with a 3D printer; (**c**) after bonding the spacer to the palatal gingival area of maxillary #3, the mouthpiece seat material is attached; (**d**) the completed mouthpiece; (**e**) the interior of a mouthpiece with attached sensors.

**Figure 6 sensors-24-00435-f006:**
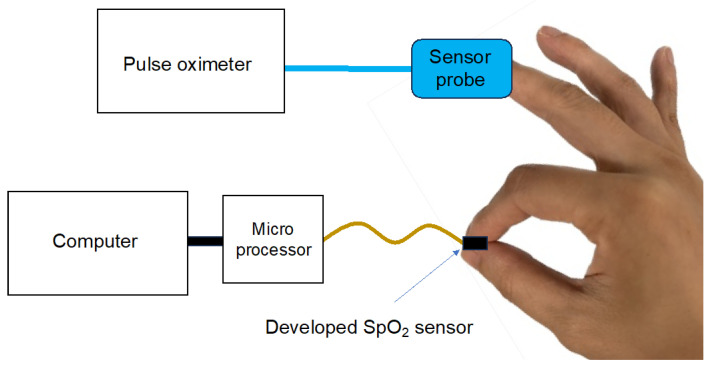
Accuracy evaluation of the developed SpO_2_ measurement device. The pulse oximeter is attached to the middle finger and the SpO_2_ sensor is grasped by the index finger.

**Figure 7 sensors-24-00435-f007:**
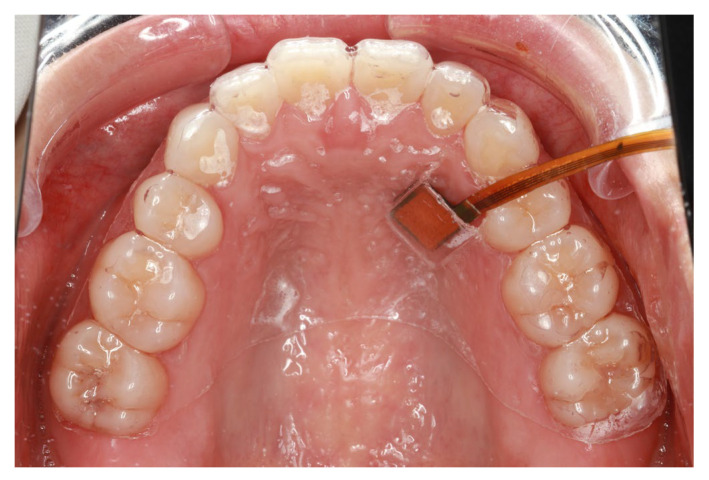
Intraoral view of the mouthpiece with the attached sensor.

**Figure 8 sensors-24-00435-f008:**
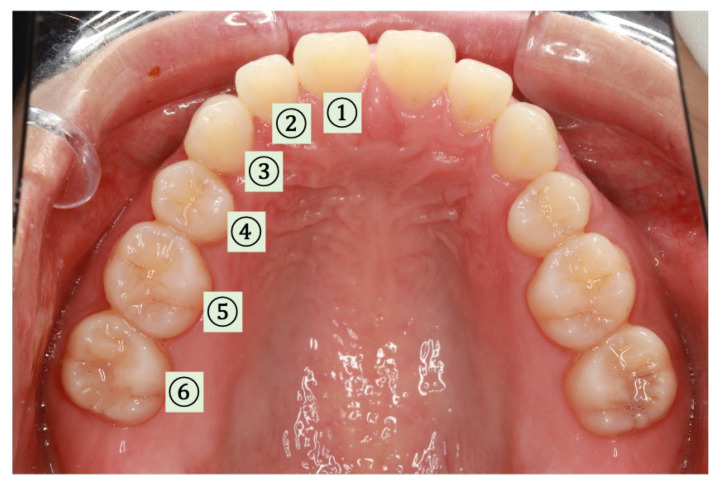
Measurement sites in the oral cavity. Sensors were placed at locations 1 to 6 to obtain the SpO_2_ and perfusion index (PI) values. Placing the sensor on the buccal side is generally painful; thus, it was only placed on the palatal side, which is less painful. (1) Central incisor. (2) Lateral incisor. (3) Canine tooth. (4) Pre-molar. (5) Lateral first molar. (6) Second molar.

**Figure 9 sensors-24-00435-f009:**
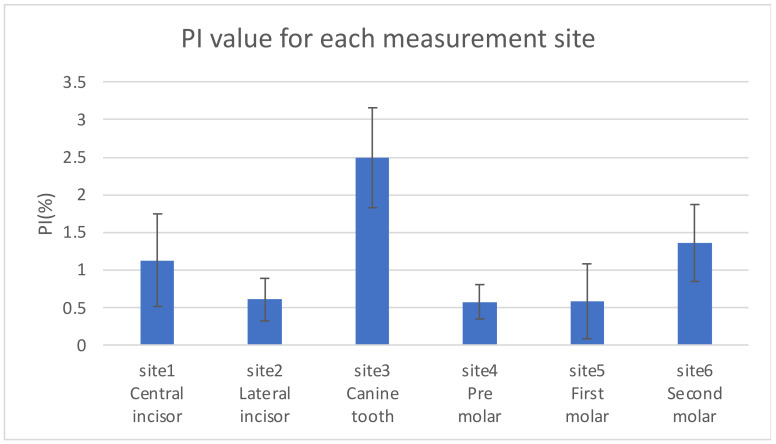
PI value for each measurement site.

**Figure 10 sensors-24-00435-f010:**
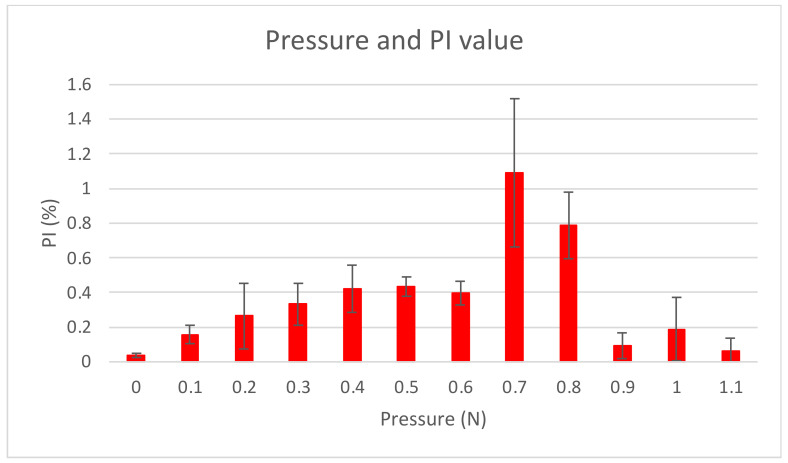
Relationship between pressure and the PI value.

**Figure 11 sensors-24-00435-f011:**
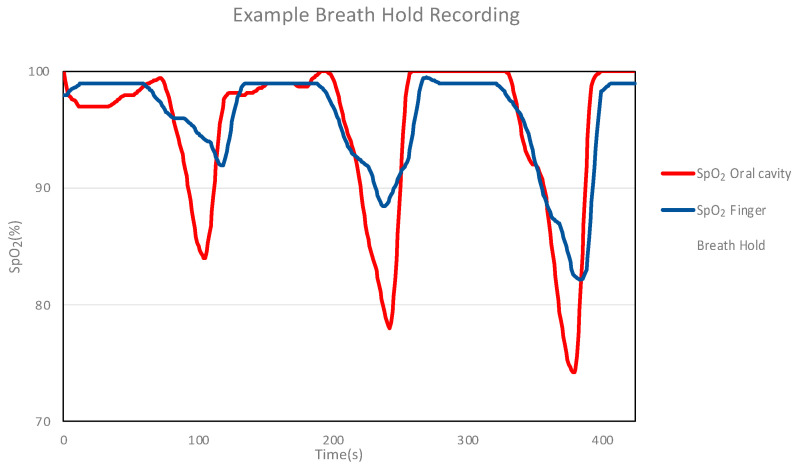
SpO_2_ values obtained from the oral cavity and finger using a mouthpiece SpO_2_ measuring device. SpO_2_ values obtained from the mouth and fingers using a mouthpiece SpO_2_ measuring device. The light blue area is the respiratory arrest zone.

**Figure 12 sensors-24-00435-f012:**
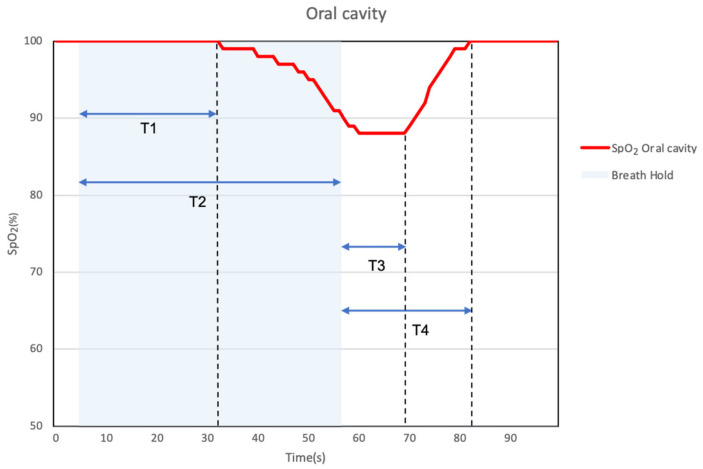
Schematic diagram of SpO_2_ values obtained using the developed SpO_2_ sensor. The time segments are classified as follows. T1: Time from respiratory arrest to the beginning of SpO_2_ decrease. T2: Time from respiratory arrest to the lowest point of SpO_2_. T3: Time from resuming respiration to the start of SpO_2_ recovery. T4: Time from resuming respiration to the completion of SpO_2_ recovery.

**Table 1 sensors-24-00435-t001:** SpO_2_ sensor condition.

LED wavelength (nm)	660 for red, 880 for IR
LED current (mA)	7 for red and IR
LED radiation power (mW)	Approx. 3.2 for red, 2.7 for IR
Sampling rate (sps)	50
Averaging (pts)	2 *^1^
LED pulse width (μs)	123.8
LED settling time (μs)	12 *^2^
Photodiode (PD) spectral bandwidth (nm)	420 to 1020 (860 nm at peak)

*^1^ A sampled PD intensity was averaged for the noise reduction to output 25 final values per second (50 sps/2 ave). *^2^ Delay from the rising edge of the LED to the start of analog–digital conversion, which allows for the LED current to be stabilized.

**Table 2 sensors-24-00435-t002:** Statistical results of T1–T4 for the mouthpiece device.

	Evaluation of Time, T1–T4		*p*-Value
	Mean Value (Second)	SD	
T1	7.4	2.8	0.01 *
T2	7.3	2.3	0.004 *
T3	7.4	1.6	<0.0001 *
T4	6.5	1.9	0.002 *

* *p*-value < 0.05 was considered significant.

**Table 3 sensors-24-00435-t003:** Statistical results for the difference in the lowest SpO_2_ values between the oral cavity and fingers.

	Mean Value (%)	SD	*p*-Value
Lowest point	4.0	0.9	0.0002 *

* *p*-value < 0.05 was considered significant.

## Data Availability

Data are contained within the article.

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
