# Peer review of "The Development of a Measuring System for Intraoral SpO2"

_sensors, 2024, doi:10.3390/s24020435_

Round 1

Reviewer 1 Report

Comments and Suggestions for Authors

The authors developed a wearable device to collect intraoral SpO2 and conducted an analysis of SpO2 measurements obtained from the fingers and oral cavity using the device. They found a notable difference in the rate of SpO2 changes between intraoral and finger measurements, including a faster response in intraoral SpO2 and a significant difference in the decrease ratio between the two measurement sites. While the potential accuracy and benefits of intraoral SpO2 over fingers (e.g., skin sensitivity, light condition, etc.) were discussed, intraoral SpO2 exhibited a significantly different reduction rate (~ 4%) while holding a breath, which may introduce significant inaccuracy, considering that subjects were healthy. The authors' discussion only explains the faster SpO2 changes in the oral cavity compared to the fingers. Overall, the research is well-written in the paper and interesting, but this concern needs to be addressed.

Author Response

To the Reviewer 1 : Thank you very much for the time and effort you put into your reviews. Your invaluable comments and suggestions helped us improve our manuscript immensely.

  1. The authors developed a wearable device to collect intraoral SpO2 and conducted an analysis of SpO2 measurements obtained from the fingers and oral cavity using the device. They found a notable difference in the rate of SpO2 changes between intraoral and finger measurements, including a faster response in intraoral SpO2 and a significant difference in the decrease ratio between the two measurement sites. While the potential accuracy and benefits of intraoral SpO2 over fingers (e.g., skin sensitivity, light condition, etc.) were discussed, intraoral SpO2 exhibited a significantly different reduction rate (~ 4%) while holding a breath, which may introduce significant inaccuracy, considering that subjects were healthy. The authors' discussion only explains the faster SpO2 changes in the oral cavity compared to the fingers. Overall, the research is well-written in the paper and interesting, but this concern needs to be addressed.

Response:

Thank you for your feedback. This study aims to enable SpO2 measurement in the oral cavity. Following a previous study (Reference 24), we have asked participants to voluntarily hold their breath for as long as comfortably possible to conveniently replicate hypoxic conditions. The significant decrease in SpO2 observed in the oral cavity compared to the finger is believed to be due to the oral cavity's proximity to the lungs, allowing blood with lower oxygen concentration to reach the measurement site more readily. This information has been added to the main text. It is particularly noteworthy that the oral cavity exhibits a faster response than the finger.

We have revised Line 329 to 332 as follows (page 12-13, lines 329-332).

・The significant decrease in SpO2 observed in the oral cavity can be attributed to the proximity of the oral cavity to the lungs compared to the fingers. It is hypothesized that blood with lower oxygen concentration reaches the measurement site more readily in the oral cavity than in the fingers.

Furthermore, we have added a note to the methods section indicating that breath-holding was performed to conveniently replicate hypoxic conditions (page 8, lines 214-215).

・Breath-holding was performed to simulate a hypoxic condition conveniently.

We sincerely thank you for highlighting this critical aspect, and we believe that these revisions will significantly strengthen the clarity and relevance of our findings. Your feedback has immensely contributed to the refinement of our research, and we are genuinely grateful for your valuable insights.

Thank you once again for your time, dedication, and thoughtful assessment of our work.

Best regards,

Reference 24. Davies, H. J.; Williams, I.; Peters, N. S.; Mandic, D. P., In-Ear Measurement of Blood Oxygen Saturation: An Ambulatory Tool Needed To Detect The Delayed Life-Threatening Hypoxaemia in COVID-19. arXiv preprint arXiv:2006.04231 2020.

Reviewer 2 Report

Comments and Suggestions for Authors

The authors demonstrated an intraoral oxygen saturation measurement device and presented some validation for the prototype. Although this work is not technologically novel, the application of known sensor technology in this paper is new and can be interesting to a wide range of biomedical engineers and medical researchers.

The following are my questions for clarifications as well as comments for consideration.

1.     Figure 9 compares PI against location of measurement. However, it was also discussed that pressure is an important parameter. Is the pressure for measurement in this comparison for each location of measurement kept the same? If not, will the pressure difference be a co-factor for the difference in PI?

2.     Figure 11 shows that compared with the standard finger-based pulse oximeter, SpO2 measurement for the presented device fluctuates observability at period where breath was not held. Can the authors explain this difference and why it does not concern the accuracy of the device?

3.     Line 260 to 268 is very confusing at initial read as it was not immediately obvious that the description of table 2 and data in table 2 is about time difference between response of the presented device and reference pulse oximeter.

4.     The writing in line 268 “was approximately 4% more sensitive” is easily misunderstood. Are the author saying that sensitivity of the presented device is 4% more sensitive compared to reference or the presented device and reference device showed a 4% difference in SpO2%

5.     Line 325 stated that the accuracy of the presented device does not differs by race of user. However, gum color is known to differ from person to person due to factors as race and even gum heath. Therefore, authors should substantiate this claim.

Thank you.

Comments on the Quality of English Language

Light editing and proofing recommended to improve clarity.  

Author Response

To the Reviewer 2 : Thank you very much for the time and effort you put into your reviews. Your invaluable comments and suggestions helped us improve our manuscript immensely.

  1. Figure 9 compares PI against location of measurement. However, it was also discussed that pressure is an important parameter. Is the pressure for measurement in this comparison for each location of measurement kept the same? If not, will the pressure difference be a co-factor for the difference in PI?

Response1:

Thank you for your valuable comments. I would like to provide a response to your inquiry regarding Figure 9. Figure 9 presents the comprehensive data of Perfusion Index (PI) values at each site (Site1 to Site6) under different pressures (0.3N, 0.4N, 0.5N, 0.6N, 0.7N, 0.8N). The attached figure includes diagrams illustrating the measured PI values at each pressure and site. Upon reviewing the attached figures, it is evident that the PI values on the palatal side of the canine are consistently the highest at every pressure level, while those on the palatal side of the premolars and first molars are consistently lower. The objective of this section is to identify the most suitable measurement site. As you rightly pointed out, variations in pressure do have a slight impact on the results at each site. However, this does not compromise the overarching observation that the palatal side of the canine consistently exhibits the highest values.

PI value for each measurement site (for each pressure)

  1. Figure 11 shows that compared with the standard finger-based pulse oximeter, SpO2 measurement for the presented device fluctuates observability at period where breath was not held. Can the authors explain this difference and why it does not concern the accuracy of the device?

Response2:

Thank you for your observation. In previous studies (conducted in the external auditory canal: Reference 24), unstable waveforms during periods of not holding one's breath have been reported. Factors affecting the stability of SpO2 include body movement, external light interference, and measurements taken immediately after sensor placement. In the current experiment, measurements were initiated approximately 30 seconds after sensor placement to account for potential instability in the measurement site. Therefore, any disruption in post-attachment measurements should have been excluded. However, during periods when breath was not held, the instability in the data measured within the oral cavity could be attributed to potential pressure from the tongue. In future measurements, it is essential to be mindful of this and exercise careful attention to ensure the tongue does not exert pressure on the sensor. For your reference, attached is a figure from measurements excluding tongue interference.

Furthermore, as a future challenge, we would like to add that there is a possibility of pressure fluctuations and instability in measurements due to tongue interference. (page 12, lines 309-312).

・However, a future challenge lies in the potential for pressure fluctuations and instability in measurements due to tongue interference. Therefore, it is essential to exercise caution during measurements to ensure that the tongue does not exert pressure on the sensor.

Figure 11 Supplementary Materials

Reference 24. Davies, H. J.; Williams, I.; Peters, N. S.; Mandic, D. P., In-Ear Measurement of Blood Oxygen Saturation: An Ambulatory Tool Needed To Detect The Delayed Life-Threatening Hypoxaemia in COVID-19. arXiv preprint arXiv:2006.04231 2020.

  1. Line 260 to 268 is very confusing at initial read as it was not immediately obvious that the description of table 2 and data in table 2 is about time difference between response of the presented device and reference pulse oximeter.

Response3:

Thank you very much for your guidance. I apologize for the confusing and misleading expression, as pointed out. Therefore, we have revised Line 270 to 279 as follows(page 11, lines 270-279).

・Table 2 summarizes the time evaluations (T1–T4) comparing oral cavity and finger measurements during breath-holding. This represents the time difference between the response measured in the oral cavity and the response measured at the finger. All T1–T4 periods showed significant differences (P < 0.05), indicating that oral cavity measurements are more responsive than finger measurements. On average, the oral cavity responded about 7 seconds (18%) earlier than the fingers across T1–T4.

・In Table 3, the results for the lowest SpO2 values are presented. Significant differences were observed (P < 0.05), with the oral cavity showing a lower SpO2 value than the finger. The lowest SpO2 value measured in the oral cavity showed a 4% difference compared to the value measured at the finger.

  1. The writing in line 268 “was approximately 4% more sensitive” is easily misunderstood. Are the author saying that sensitivity of the presented device is 4% more sensitive compared to reference or the presented device and reference device showed a 4% difference in SpO2%

Response4:

Thank you for your feedback. I apologize for the confusing English expression. The device used for measurements is the same for both the oral cavity and the fingers. To convey this more clearly, the main text has been revised and additional information has been included as follows.(page 8, lines 214-215).

・The device used for measurements is the same for both the oral cavity and the fingers.

We have revised Line 278 to 279 as follows(page 11, lines 278-279).

・The lowest SpO2 value measured in the oral cavity showed a 4% difference compared to the value measured at the finger.

  1. Line 325 stated that the accuracy of the presented device does not differs by race of user. However, gum color is known to differ from person to person due to factors as race and even gum heath. Therefore, authors should substantiate this claim.

Response5:

Thank you very much for your valuable comments. I appreciate your insights regarding the gingival pigmentation, and I would like to provide an explanation.

The color of the gums varies depending on factors such as race, and there is significant individual variation. Some individuals have less melanin in the gums compared to the skin, while others may exhibit the opposite. The amount of melanin in the oral mucosa can influence the measurement of SpO2, potentially resulting in readings higher than the actual blood oxygen saturation, similar to the effect observed in the skin (Please refer to the following references.).

As you rightly pointed out, there is a potential lack of accuracy in the mentioned section, and I am inclined to remove it to ensure precision. In making these revisions, I will be careful to maintain the overall coherence of the research and strive to convey a clearer argument.

If you have any further questions or concerns, please feel free to let me know.

Thank you for your valuable feedback.

Reference. Mirdad, A.; Alqarni, M.; Bukhari, A.; Alaqeely, R., Gingival Pigmentation Features in Correlation with Tooth and Skin Shades: A Cross-Sectional Study in a Saudi Population. Oral Health & Preventive Dentistry 2023, 21, (1), 285-290.

Round 2

Reviewer 2 Report

Comments and Suggestions for Authors

Thank you for the revision. Last comment from me as below. 

Please confirm that arrow between ADC and Controller is in the correct direction for figure 2.

Author Response

To Reviewer 2;I sincerely appreciate your thoughtful comments and guidance. Your meticulous review has been invaluable, and I am grateful for the careful consideration you gave to our manuscript.

 Please confirm that arrow between ADC and Controller is in the correct direction for figure 2.

Response:

In response to your observations, we have corrected the issue with the arrow direction in Figure 2. I apologize for any oversight on our part and sincerely thank you for bringing it to our attention.

The revised figure is attached for your review. We kindly ask for your confirmation, and we hope these changes meet your expectations and effectively address the points you highlighted.

Fig2 revised

Thank you once again for your thorough review and invaluable insights. Your contribution to the peer review process is truly commendable.

Best regards,

Yuki Kashima, D.D.S.